# Overview of the Potential Beneficial Effects of Carotenoids on Consumer Health and Well-Being

**DOI:** 10.3390/antiox12051069

**Published:** 2023-05-10

**Authors:** Pasquale Crupi, Maria Felicia Faienza, Muhammad Yasir Naeem, Filomena Corbo, Maria Lisa Clodoveo, Marilena Muraglia

**Affiliations:** 1Interdisciplinary Department of Medicine, University of Bari “Aldo Moro”, 70125 Bari, Italy; pasquale.crupi@uniba.it (P.C.); marialisa.clodoveo@uniba.it (M.L.C.); 2Department of Precision and Regenerative Medicine and Ionian Area, University of Bari, “Aldo Moro”, 70124 Bari, Italy; 3Department of Plant Production and Technologies, Faculty of Agricultural Sciences and Technologies, Nigde Omer Halisdemir University, Nigde 51240, Turkey; naeem.yasir91@yahoo.it; 4Department of Pharmacy-Drug Sciences, University of Bari “Aldo Moro”, 70125 Bari, Italy; filomena.corbo@uniba.it (F.C.); marilena.muraglia@uniba.it (M.M.)

**Keywords:** lutein, lycopene, β-carotene, human health, antioxidant, food supplements, chronic diseases

## Abstract

Well-known experimental research demonstrates that oxidative stress is the leading cause of the onset and progression of major human health disorders such as cardiovascular, neurological, metabolic, and cancer diseases. A high concentration of reactive oxygen species (ROS) and nitrogen species leads to damage of proteins, lipids, and DNA associated with susceptibility to chronic human degenerative disorders. Biological and pharmaceutical investigations have recently focused on exploring both oxidative stress and its defense mechanisms to manage health disorders. Therefore, in recent years there has been considerable interest in bioactive food plant compounds as naturally occurring antioxidant sources able to prevent, reverse, and/or reduce susceptibility to chronic disease. To contribute to this research aim, herein, we reviewed the beneficial effects of carotenoids on human health. Carotenoids are bioactive compounds widely existing in natural fruits and vegetables. Increasing research has confirmed that carotenoids have various biological activities, such as antioxidant, anti-tumor, anti-diabetic, anti-aging, and anti-inflammatory activities. This paper presents an overview of the latest research progress on the biochemistry and preventative and therapeutic benefits of carotenoids, particularly lycopene, in promoting human health. This review could be a starting point for improving the research and investigation of carotenoids as possible ingredients of functional health foods and nutraceuticals in the fields of healthy products, cosmetics, medicine, and the chemical industry.

## 1. Introduction

Phytonutrients, also known as phytochemicals, are biologically active compounds made by plants that have attracted interest for their potential applications in food, medicine, and cosmetics [1,2,3]. Numerous plant compounds can be helpful in the treatment of a variety of serious diseases because they reduce oxidative stress [4]. In this scenario, carotenoids make a significant contribution to human health, particularly in relation to cancer and cardiovascular disorders (CVDs) [1]. Carotenoids such as lutein and zeaxanthin are known to be useful for bone health, while lycopene is known to be advantageous for prostate cancer and other health benefits [1].

Carotenoids are naturally occurring bright-colored compounds which perform critical biological changes in all photosynthetic organisms such as algae, cyanobacteria, and plants [5]. Carotenoids are found in chloroplasts in green plant tissues (leaves, stems, seeds, and unripe fruits) and are linked with chlorophyll for absorbing light energy at specific wavelengths [6,7]. They protect the photosynthetic apparatus, dissipate excess light energy, and reduce the reactivity of harmful species such as “singlet” oxygen and the excited (i.e., higher energy state) chlorophyll; in addition, they shield plant cells against superoxide radicals and light damage. Carotenoids add color, varying from yellow to red, to fruits and flowers while aiding in pollination and seed dissemination via vector interest [8]. It is worth noting that carotenoids cannot be biosynthesized by a great number of animals, while they may be ingested by the diet [9]. Since the 1980s, there has been a significant upsurge in carotenoids as putative health-promoting substances. Therefore, adequate carotenoid consumption could be linked to lower hazards of heart disease, bone, skin, and eye conditions, as well as several malignancies (including cervical, ovarian, colorectal, prostate, and breast) [10]. According to Melendez-Martınez [6], fruits and vegetables constitute the most significant sources of carotenoids in the human diet. However, a large quantity of carotenoids can be found in seafood, milk products (cheese, butter, etc.), egg yolks, and other foods originating from animals [8]. Surprisingly, lutein and some other carotenoids are also found in babies’ diets since they are released in the mothers’ breastmilk. Numerous studies were focused on determining the carotenoid levels in foods, and the gathered information has been collated in databases [11]. In addition to selecting, breeding, and improving classic cultivars of well-known staple foods, there is growing interest in finding novel natural sources of carotenoids, such as underused wild fruits and vegetables [12].

In this review, we have focused our attention on the ability of carotenoids, particularly lycopene, to reduce oxidative stress. Strong attention was paid to their potential exploitation in the treatment of inflammatory, diabetic, and cardiovascular pathologies in order to suggest carotenoid biomolecules as new high-value ingredients for health products (nutraceuticals, pharmaceuticals, and cosmeceuticals).

## 2. Biochemistry of Carotenoids

There are over 600 carotenoids in nature, the majority of which are colored and synthesized by bacteria, fungi, and plants. Their distinctive characteristic is an extensive conjugated double-bond system, which is able to absorb light in an inclusive range between 400 and 550 nm but is also responsible for the instability of these compounds. For example, they are easily oxidized when exposed to the air and light temperature, and acids can affect the double bonds in the natural *trans* configuration of carotenoids to form several *cis*–*trans* stereoisomers [13]. Food carotenoids are usually C_40_ tetraterpenoids with a fundamental supporting structure of eight isoprenoid units [14]. The basic linear and symmetrical skeleton of their molecule consists of a central position, with 22 C, and two terminal units of 9 C atoms. These terminal units can be acyclic (i.e., lycopene) or can be cyclized at one or both ends (i.e., α, β-carotene or γ-carotene). In the latter case, oxygenated functional groups can also be attached, giving rise to alcohols, ketones, esters, or epoxide structures [15,16].

Carotenoids are often called by their biological origin, such as carotene from carrot, zeaxanthin from *Zea mays*, and lutein from *Macula lutea*. However, to facilitate discussion about these important molecules, a systematic approach to nomenclature has been defined [17]. Traditionally, carotenoids are classified into two main structural groups: (a) carotenes, characterized by the presence of only C and H atoms, and (b) xanthophylls, presenting different oxygenated functional groups in addition to C and H (Figure 1) [13].

Among the carotenes, β-carotene is the most widespread, followed by α-carotene and γ-carotene present in lower concentrations (e.g., in apricots, mangos, cherries, carrots, and grapes). Of the acyclic carotenes, lycopene (the principal pigment of many red-fleshed fruits, such as watermelon and tomato) is the most common. As regards xanthophylls, (all-*trans*)-lutein is the most widespread, whereas smaller amounts of zeaxanthin and epoxyxanthophylls, neoxanthin, violaxanthin, luteoxanthin, and lutein-5,6-epoxide can also be present, especially in green vegetables [11].

## 3. Main Sources of Carotenoids in the Diet

The three hydrocarbon carotenes: α-carotene, β-carotene, and lycopene, as well as the three oxygenated xanthophylls: lutein, zeaxanthin, and β-cryptoxanthin, are the most researched carotenoids in diets. There has recently also been considerable interest in colorless carotenoids, such as phytoene and phytofluene, since they are very abundant in the diet and might give health and beauty advantages [11]. The amount of carotenoid in food may be categorized into four distinct dosage groups: low (0–0.1 mg/100 g), moderate (0.1–0.5 mg/100 g), high (0.5–2 mg/100), or extremely high (>2 mg/100 g). It is crucial to mention that the level in food items can be affected by many factors, such as genotype, climate, agronomic practices, cooking, processing, and preservation techniques [18]. The amounts of pro-vitamin A carotenoids in fruits and vegetables have subsequently been found to be positively impacted by climate change, potentially because of changes in the Sun UV index [19]. Principally fruit and vegetables, but also cereals and foods of animal origin, are important sources of carotenoids, suggesting that a varied diet is a good choice for the right intake of these compounds.

### 3.1. Fruits and Vegetables

Fruits and vegetables are the most significant food sources of carotenoids for humans. Frequently grown and consumed plants and fruits (green vegetables, carrots, red peppers, tomatoes, apricots, peaches, mango, papaya, and other citrus species) are good contributors of carotenoids [13,18,20] (Figure 2).

The most prevalent and significant pro-vitamin A carotenoid is β-carotene, which is abundant in orange and yellow vegetables such as carrots and some types of pepper, as well as dark green leafy vegetables such as kale, spinach, common purslane, and lettuce [21,22]. The vegetables that most contribute to the intake of carotenoids in the diet include spinach (128.1 µg/day), tomato (299 µg/day), and carrot (573 µg/day), both raw and cooked, while, regarding the fruits, these are tangerine (15.3 mg/day), orange (12 mg/day), and banana (11.2 mg/day) [23].

Lycopene is a red pigment giving tomato, papaya, apricot, pink guavas, and watermelon their reddish color [18,24]. It is also present in several plants that are neither red nor orange, such as parsley and asparagus [25]. The most widely used sources of lycopene are tomatoes and products made from tomatoes [26] with concentrations varying based on the cultivar, climate, geographic origin, and processing techniques [27]. However, it was discovered that Rosa mosqueta fruits have more lycopene (392 mg/kg d.w.) than tomato [28]. In the pericarp and pulp of completely ripe pink guavas, (all-E)-lycopene and (15Z)-lycopene account for 280.5 and 291.4 g mg/kg d.w. [29].

Green leafy vegetables are the major source of lutein in the human diet; they may be found in spinach, common purslane, kale, watercress, broccoli, Brussels sprouts, parsley, and lettuce [22,30,31]. Orange and red peppers are common important sources of zeaxanthin; however, other research has reported as the concentration of this xanthophyll is very high in goji berries (35.7 mg/100 g f.w.) and Romanian sea buckthorn (19.3–42.4 mg/100 g d.w.) [20,32].

### 3.2. Cereals and Their Products

In cereal grains, carotenoids such as lutein, zeaxanthin, β-cryptoxanthin, and α- and β-carotene can be detected [33]. Carotenoid concentration in maize products varies according to the variety and/or processing, as demonstrated by canned maize (17.53–27.94 mg/g f.w.). Yellow maize has historically been regarded as the sole cereal with a significant carotenoid concentration; the total carotenoid content in maize grain (11.14 mg/g d.w.) is approximately thirty times greater than that of oats, wheat, or barley (0.36, 1.50–3.05, and 1.50 mg/g d.w., respectively). Furthermore, the same researchers noted that cornflakes and meal had roughly similar carotenoid concentrations (15.10–21.28 and 16.37–19.33 mg/g f.w., respectively) which were greater than in flour (8.25–19.20 mg/g f.w.). Carotenoid concentration is further reduced by processing: yellow maize tortillas and chips have 2.13 and 1.42 mg/g d.w., respectively [33].

Bread wheat (*Triticum aestivum*), a globally basic grain, represents the most frequently farmed wheat species. Due to the presence of lutein, the endosperm of wheat grains and products made from wheat (mostly flour and baked foods) have a yellowish hue. In most wheat species, lutein accounts for more than 85% of the total carotenoid concentration. Bread and pasta contain varying amounts of lutein and zeaxanthin, corresponding to 4.5–6.3 and 0.08–0.12 mg/g d.w., respectively [33]. Additionally, if pasta is made with eggs, an even greater carotenoid concentration can be anticipated: 8.50 mg/g d.w. for total carotenoid content, which includes 6.56 mg/g d.w. of lutein and 1.61 mg/g d.w. of zeaxanthin [34]. Cooked rice is not a significant source of carotenoids in human diets, in contrast to goods made from wheat. The carotenoid content of parboiled rice is significantly reduced during processing. Zeaxanthin (14–37 ng/g f.w.) follows lutein (91–107 ng/g f.w.) and β-carotene (66–150 ng/g f.w.) as the main carotenoid in untreated parboiled brown rice [35].

### 3.3. Dairy Goods

Carotenoids (especially β-carotene and lutein) play a role in the sensory qualities of dairy foods despite having a low proportion in milk. β-carotene concentration affects the yellow color of butter and various kinds of cheese while cheese making results in significant retinol losses [36]. However, not all the carotenoids found in dairy items come from milk. *Thermus thermophilus*, a genus that produces carotenoids, was recently shown to be present at greater levels in pink cheeses. The pinking of cheeses was subsequently replicated by adding a *T. thermophilus* isolated from a testing cheese during the production process [37]. Additionally, during production, dairy products receive a direct addition of carotenoids. Traditional methods for coloring cheese include adding annatto, a long-used pigment made from the outer coating of the seeds of the tiny tropical tree Bixa orellana, which contains the apocarotenoids bixin and norbixin [38].

### 3.4. Fish

Aquafeeds cause the presence of carotenoids in fish, influencing the color [39]. Fish have a wide range of carotenoids, although xanthophylls are known to be more absorbed and accumulated than carotenes [11]. Zeaxanthin, astaxanthin, tunaxanthin, and lutein are the most prevalent xanthophylls, and they accumulate in tissues such as muscle, integuments, liver, eggs, gonads, eyes, brain, gut, and mouth mucus. Trout muscles, for instance, have been discovered to contain carotenoids such as lutein, zeaxanthin, canthaxanthin, β-cryptoxanthin, and astaxanthin [40].

### 3.5. Livestock

Mammals can be divided into two categories based on the accumulation in adipose tissue. Animals with white fat, which absorbs little to no carotenoids, fall into one category, while those with yellow fat, which can absorb carotenoids, fall into the other. The first category includes pigs, goats, sheep, and rodents, while the second category includes cattle, horses, and birds [11]. Thus, the color of bovine fat is attributable to the presence of β-carotene [41], the most abundant carotenoid, as well as other pigments such as lutein. According to research using the muscle and liver of numerous meat-producing species, about 0.6 mg of carotenoids per day could be obtained from eating cow or horse liver [42]. The scientific community has made several efforts to establish how animal food has a direct influence on the content of carotenoids and to discover optimal feeding methods to raise this level [43,44,45]. The major carotenoid found in bovine serum and adipose tissues is β-carotene.

### 3.6. Eggs

The capacity of hens to absorb carotenoids from their food and deposit them in eggs is the cause of egg yolk coloration [46]. Therefore, poultry on a diet consisting of maize and soybean meal will produce eggs containing lutein, zeaxanthin, β-cryptoxanthin, and β-carotene. Eggs produced from a diet comprising 60% maize had 14.2, 5.7, 1.3, and 1.4 mg/g of these carotenoids, respectively. The two main carotenoids found in egg yolks are lutein and zeaxanthin, though their concentrations can vary greatly [47]. The yolk color is the main determinant of consumer product approval, as well as the discriminant factor for the technological application in pasta production or bakery products, and the required pigmentation is typically obtained by supplementing diets with carotenoids, particularly when low-carotenoid feeds are utilized. The carotenoid content of eggs is reduced during boiling and processing as they are not often eaten uncooked. Cooking reduced zeaxanthin by 8% to 15.2% and canthaxanthin by 11.3% to 12.8%, with lutein being the carotenoid most impacted by processing (22.5% decrease after boiling, 16.7% decrease after microwaving, and 19.3% decrease after frying) [46,47].

### 3.7. Alternative Sources

In order to support a rising population with sustainable and healthful meals, the agro-food system is undergoing a significant shift that must happen quickly [48]. Sustainability and the circular economy are two ideas that ought to always be connected to food production. Research into “alternative” sources of carotenoids (e.g., macroalgae) is becoming more significant in this situation [49]. Fungi, bacteria, or insects are additional sources of carotenoids that can be further explored because of their features and production benefits, such as diversity, reduced resource consumption, and the ability to optimize growing conditions for various purposes [50]. The content of antioxidant compounds, together with the presence of high-biological-value proteins, could also represent an additional argument in favor of the consumption of non-traditional but environmentally friendly food (such as insects) in Western countries.

## 4. Biological Availability and Accessibility of Carotenoids

More important than the quantity of the meal are the contents of bioactive and nutritional components which are solubilized for digestibility (referred to as the bioaccessible fraction) [51]. The large amount of scientific evidence confirms that carotenoids are linked to proteins and certain other large molecules in the chloroplast of cells, which are surrounded by hard cell walls that serve as structural inequalities for the release of carotenoids [52]. Additionally, as carotenoids are lipid-soluble, the amount of lipids in a diet is crucial for their bioaccessibility. Then, the microstructural characteristics of food, such as pectin, a significant nutritional fiber found in fruits and vegetables that affects carotenoid bioaccessibility, have an impact on lipid digestion [51,52]. The absorption of carotenoids may also be impacted by dietary phytochemicals such as polyphenols, phytosterols, fatty acids, tocopherols, and divalent metals [53].

The stability and bioavailability of carotenoids are strongly influenced by their molecular structure [54]. In comparison to astaxanthin monoester and free astaxanthin, astaxanthin diesters containing long-chain and saturated fatty acids (SFAs) showed greater thermal stability (at 60 °C) [54]. As unsaturated fatty acids (UFAs) are more likely to oxidize than SFAs, it is possible that increasing the length of the carbon chain, reducing the unsaturation of fatty acids, and increasing the esterification degree of astaxanthin (diesters and monoesters) were all advantageous to the compound thermal stability. The astaxanthin concentration in mice serum revealed that astaxanthin esters with short-chain fatty acids (SCFAs) had higher bioavailability than long-chain fatty acids (LCFAs), while astaxanthin esters with high UFAs had greater efficacy than those with SFAs [15]. It was also interestingly demonstrated that the bioavailability of astaxanthin showed opposite trends to thermal stability. Astaxanthin esters with SCFAs are likely hydrolyzed to free astaxanthin by cholesterol ester hydrolase (CEH) before being inactively absorbed through the brush border of the enterocytes, which suggests that astaxanthin esters with SCFAs can be more easily digested into free astaxanthin than astaxanthin with LCFAs [55]. Astaxanthin monoesters showed a much better bioavailability than astaxanthin diesters, in contrast to the thermal stability examined. Astaxanthin-DHA monoester had the best bioavailability among all astaxanthin structures examined [15].

Like other carotenoids, lycopene is mixed with fatty acids and bile acids in a complex after food is consumed and released in the duodenum. Lycopene-containing micelles have been created as a result. Such micelles have hydrophilic shells with a hydrophobic center that contains lycopene [29,55]. Afterward, lycopene is digested and packed into chylomicrons in the small intestine by passively regulated diffusion [52]. It is well known that regular intake of lycopene can prevent chronic diseases such as cardiovascular disease, type 2 diabetes, high blood pressure, kidney disease, bone, skin and eye diseases, and cancer. However, thermal processing, light, oxygen, and enzymes in the gastrointestinal tract (GIT) impair the bioaccessibility and bioavailability of ingested lycopene throughout the diet [56]. Several literature studies suggest nanoencapsulation as a potential platform to protect lycopene from external physical agents and enzymatic activity of the human digestive system, favoring its bioaccessibility and use as a functional food ingredient for therapeutic treatments [57]. Then, it is secreted into the bloodstream and lymphatic system before being transferred to the liver and other bodily organs, as shown in Figure 3 [30]. Lycopene is one of the carotenoids known to accumulate in high concentrations in the adrenal glands, testes, and prostate [28,55]. Additionally, a smaller proportion of it might be detected in other body areas such as the brain and skin [58].

Moreover, a variety of lycopene isomers, due to physiological *cis–trans* isomerization occurring in the intestinal epithelium, liver, and stomach, may be found in human plasma [59]. Both *trans* and *cis* isomers, accounting for up to 50% of the total amount of lycopene, can be cleaved eccentrically by β-carotene oxygenase 2 (BCO2) to give rise lycopenals, lycopenols, and lycopenoic acids, whose beneficial effects were recently highlighted [52].

## 5. Role of Carotenoids in Human Health and Disease: The Lycopene Case

Any substance that raises stress levels is a stressor. Stress is the overall reaction of an organism to adverse circumstances. Stress disrupts physiological homeostasis by eliciting a biological response to external stimuli [60]. One of the fundamental causes of chronic disease is oxidative stress, which is caused by extremely reactive free radicals. Antioxidants are a diverse group of compounds that suppress oxidation in various ways [61]. In this sense, carotenoid consumption reduces the risk of a variety of chronic illnesses, including CVDs and neurological disorders, type 2 diabetes, and different types of cancer [62,63]. Pro-inflammatory mediators such as oxidized phospholipids (OxLDLs), circulating pro-inflammatory cytokines (interleukin-8, -6, and -1), inflammatory-stimulating prostaglandin E2 (PGE2), tumor necrosis factor-alpha (TNF-alpha), and nuclear factor kappa-light-chain-enhancer of activated B cells (NF-κB) are generally associated with higher levels of C-reactive protein (CRP). Due to their antioxidant capabilities, carotenoids can modulate oxidative stress or cause an upregulation of antioxidant and cytoprotective phase II enzymes via nuclear factor erythroid 2-related factor 2 (Nrf2) and peroxisome proliferator-activated receptor (PPAR) [64,65]. The primary function of carotenoid-mediated Nrf2 signaling is to reduce oxidative stress and inflammatory reactions [66,67].

In particular, lycopene, because of its lengthy chain of conjugated double bonds, is the most effective free radical and single oxygen scavenger among the 600 naturally occurring carotenoids. It can prevent lipid oxidation through its initial phases since it is a potent singlet oxygen detoxifier. It has been demonstrated that lycopene can defend against oxidative stress more effectively than other carotenoids, including β-cryptoxanthin, β-carotene, lutein, and zeaxanthin [68]. The primary biological function of lycopene is to shield DNA from oxidative stress in order to avoid mutations that might cause chronic illnesses [69]. Cell viability, the immune system, and gene transcription are all impacted by its modulation of phase I and phase II detoxifying enzymes. The antioxidant response element (ARE) transcription system is associated with the *cis*-regulatory sections of detoxifying enzyme promoters. Lycopene can influence xenobiotic metabolism by breaking the cytosolic connections between the main ARE-activating Nrf2 and its inhibitor (Keap1) via activating the ARE transcription pathway [70]. Lycopene generates ROS by three mechanisms: radical addition (adduct production), electron transfer to the radical, and allylic hydrogen abstraction [71].

The production of acylation and allylic hydrogen abstraction are two mechanisms that contribute to lycopene’s antioxidative impact. These possible interactions are influenced by the kind of responding free radical, the structural properties of lycopene, and the location and orientation of lycopene inside the membrane in biological systems [72].

There are several methods by which lycopene and free radical events can happen simultaneously. By recovering non-enzymatic antioxidants such as vitamins E and C from their radicals, lycopene can strengthen the cellular antioxidant defense system [71]. When a system produces singlet oxygen, lycopene acts as an antioxidant; yet, when a system produces peroxide, lycopene acts as a pro-oxidant. The redox potential of lycopene makes it a useful antioxidant [73]. Furthermore, lycopene exhibits pro-oxidant behavior at high levels but antioxidant behavior at low amounts. Pro-oxidant efficacy is influenced by a variety of variables, such as tissue oxygen tension, lycopene content, and associations with several other antioxidants. Lycopene may affect biological systems in both positive and negative ways as a pro-oxidant, and it may also have an impact on how diseases develop in people. Lycopene may aid in preventing the development and spread of malignant lesions, as well as tumor cytotoxicity if it functions as a pro-oxidant in already injured cells. Antioxidant connections in carotenoids can reduce their pro-oxidant effects, strengthening their antioxidant properties [74].

There are several in vitro and ex vivo studies on cultured cells and animal models, respectively, reporting the beneficial effects of carotenoids in glycemic and lipidic impairment, anti-inflammatory status, and tumor cell apoptosis and proliferation. Carotenoids can be utilized for the treatment and prevention of diabetes in terms of animal research and epidemiological investigations. Lycopene reduced blood and urine glucose levels, as well as diabetes-related pancreatic damage in a diabetic rat model (STZ-induced). The serum insulin levels were also raised [75]. In diabetic Wistar rats, lycopene treatment dramatically decreased serum nitrate–nitrite levels [76]. Lycopene also significantly reduced hepatic glycogen levels and other HFD-related changes in blood sugar, insulin, fasting blood sugar, and insulin intolerance. Lycopene ingestion significantly reduces STAT3 expression and phosphorylation in the livers of HFD mice. Additionally, an elevation in STAT3 activities caused by an adenovirus significantly halted the decline in fasting blood glucose and insulin levels [77]. It is interesting to note that CD-1 male mice with cognitive impairments have better D-galactose after consuming lycopene (50 mg/kg BW/day). Additionally, it dramatically decreased the levels of malondialdehyde (MDA), total sialic acid, DNA fragmentation, and antioxidant enzyme activity in a rat model of colitis [71]. HDL levels considerably rose while LDL, triglycerides, and total cholesterol levels fell in rats given lycopene supplementation. Both a considerable drop in TG and a decrease in oxidized LDL have been seen in lycopene-supplemented rats and hamsters, respectively [76].

In carrageenan-induced inflammation, lycopene administration (12.5 mg/kg BW) significantly reduced edema in Swiss mice, investigated by a variety of phlogistic substances and immunostaining for COX-2, iNOS, and NF-B [68]. In recent research, Sprague Dawley rats were given a single 200 g injection of LPS to produce endotoxin-induced uveitis [62]. Additionally, lycopene is effective in treating rat brain mitochondrial dysfunction brought on by A 1–42 in conjunction with TGF, elevated pro-inflammatory cytokines, TNF-alpha, and IL-1, as well as NF-κB and caspase-3 activity. The levels of vascular cell adhesion molecule 1 (VCAM-1) and ICAM-1 were considerably reduced by lycopene from tomato juice [68]. By lowering the levels of IL-6 and TNF-alpha, inhibiting MAPK activity, and blocking the NF-κB transcription factor, lycopene also shields rats against acute lung damage brought on by LPS. The consequence is the restoration of normal metabolism [78]. Furthermore, a study using hairless mice demonstrated that lycopene could alleviate atopic dermatitis (AD) symptoms such as visual cues, skin moisture levels, inflammatory cells in the dermis, and skin thickness following administration [79].

Human colorectal cancer cells were treated with lycopene dosages of 0, 10, 20, and 30 M. Using the 3-(4,5-dimethylthiazol-2-yl)-2,5-diphenyl tetrazolium bromide (MTT) technique, the effect of lycopene on cell proliferation was examined. After lycopene delivery, levels of NO and PGE2 decreased [80]. Lycopene extracts (5 mg/mL) were used to treat human PCa cells, and the results showed a substantial decrease in cell viability and apoptotic cell population [81]. Additionally, lycopene nanoparticles (in lipid-based wall material in MCF-7 cells) drastically decreased cell viability and survival in a time- and concentration-dependent manner. The danger of human breast cancer cells was decreased by lycopene extract (dose: 400 and 800 g/mL) (MCF-7). It may alter the size and granularity of the MCF-7 cell, as well as the potential of the mitochondrial membrane and DNA fragmentation. The cell membrane is also not noticeably damaged, and regular apoptosis and necrosis do not occur [82]. Lycopene consumption increased the pro-apoptotic X protein-linked B cell lymphoma gene and inhibited the mitogen-activated and B-protein kinase signaling pathways [78]. Consuming lycopene significantly reduced the metastatic burden in a tumor-bearing rat model of ovarian cancer, as well as the expression of related indicators such as CA125 [80]. Supplementation (200 to 400 mg lycopene/kg diet) in laying hens dramatically decreased the incidence of ovarian tumors, as well as the size and quantity of the tumors. Additionally, lycopene administration decreased the expression of STAT3 in ovarian tissues by activating the protein inhibitor of activated STAT3 expression [83].

However, considering the public interest in the preventive and therapeutic action of food constituents on human health and disease, recent evidence of human studies relating to carotenoids’ effects against chronic inflammation and non-communicable diseases has been revised.

### 5.1. Cardiovascular Diseases

The main disease and fatality factor worldwide is CVD. Smoking, high cholesterol, and blood pressure are all significant risk factors for coronary disease. Atherosclerosis is the most common cause of CVDs, which impairs the heart and brain. They are caused by damage to and remodeling of blood arteries, and they obstruct blood flow [84]. The major cause of stroke, heart attacks, renal failure, and several other problems is thought to be hypertension [85]. Positive correlations between a greater intake of carotenoids and a reduced risk of CVDs have been shown in scientific studies [86]. In particular, lycopene intake significantly lowered systolic blood pressure (SBP), notably in individuals with an initial SBP of 130 mmHg [85]. Another piece of evidence from the USA and Finland exhibited that maximum supplementation of lycopene with an average of 9.81 mg daily can lower the risk of stroke [86]. Lycopene is a heart-protective nutritional supplement. It has been demonstrated that blood lycopene levels lower the risk of major cardiovascular problems [87]. Lycopene’s preventative impact on cardiovascular disease is extensively supported by epidemiological research; it can remove some of the strong oxidants known to be linked to atherosclerosis and lessen the oxidation of cholesterol. Supplementing with lycopene has been demonstrated to raise blood lycopene levels, lower oxidative stress indicators, and enhance antioxidant status [26]. Additionally, it was shown that lycopene protects transplanted arteries by controlling the production of vital proteins required for the development of arteriosclerosis. The activity of ROCK1, Ki-67, ICAM-1, and ROCK2 was noticeably reduced, whilst the activation of eNOS-implanted arteries and plasma cGMP levels were enhanced. Monocyte–endothelial cell contact and NF-κB activation caused by TNF-alpha were reduced by lycopene [87]. Serum lycopene was discovered to be negatively correlated with VCAM-1 and LDL [88]. Pre-clinical investigations have shown that lycopene consumption can enhance endothelial cell function. Through its antioxidant action, lycopene has the power to enhance NO bioavailability and endothelium-regulated vasodilation, lessen protein, DNA, and lipid damage, and enhance mitochondrial function [88].

Clinical studies have shown that lycopene and tomato products lower total cholesterol and low-density lipoprotein cholesterol (LDL-C) [89]. Supplementing with lycopene can help healthy post-menopausal women, and might lower LDL and total cholesterol. Lycopene may reduce the expression of Rho-related kinases and can also regulate the expression of the NO/cGMP pathways to alleviate vascular arteriosclerosis following allograft transplantation [76].

### 5.2. Type 2 Diabetes

Type 2 diabetes (T2D) is a chronic metabolic disease characterized by high blood glucose levels, causing serious damage to the cardiovascular, renal, and respiratory, as well as other, systems. The global diabetes prevalence and health expenditure are currently rising and are expected to reach 700 million people and USD 776 billion by 2045. A healthy lifestyle, primarily including the adoption of a healthy diet rich in fruit and vegetables, is the cornerstone for the glycemic control and prevention of TD2 [90]. Recent meta-analysis studies have examined the associations of dietary intakes and circulating concentrations of carotenoids with the risk of T2D. Higher dietary intakes and circulating concentrations of total carotenoids, especially β-carotene, have been associated with a lower risk of T2D. Furthermore, lycopene intake has exerted a fasting blood glucose (FBG) decreasing effect. However, despite the promising results, the same authors warn about the need to carry out more studies in order to confirm the causality and explore the role of foods rich in carotenoids in the prevention of T2D [90,91].

The use of lycopene as a complementary medicine for T2D is limited and controversial. Leh et al. evaluated the effect of lycopene intake on the changes in glycemic status (measured as glycated hemoglobin and FBG) and antioxidant capacity among 87 T2D patients versus 122 healthy individuals. Direct positive correlations were found between lycopene intake and peripheral antioxidant levels among T2D patients. Contrarily, glycated hemoglobin and FBG levels decreased significantly with the higher lycopene intake. Overall, this case–control study showed how lycopene may act to ameliorate oxidative stress and improve the pathophysiology of T2D [92].

### 5.3. Cancer

Carotenoids may have a favorable effect in slowing the development and spread of cancer, according to a number of epidemiological, clinical, and pre-clinical investigations [63,93]. However, a recent expert assessment from the World Cancer Research Fund (WCRF) and American Institute for Cancer Research (AICR) found that there is only minimal or insufficient data to draw a conclusion that eating food containing carotenoids lowers the chance of developing colon cancer [94]. Furthermore, there is an indication of a substantial inverse association between carotenoid consumption and the risk of developing ER-negative breast cancer. An evaluation of 6 cohort, 11 case–control, 3 cross-sectional, and 2 controlled clinical trials on the effect of carotenoids on the incidence of prostate cancer suggested higher intakes of carotenoids, particularly lycopene from tomatoes, may be associated with a lower risk of developing the disease [95]. Intake of α-carotene was substantially linked to improved breast cancer survival in a comprehensive review and meta-analysis of 8 cohorts, 1 pooled study, 1 clinical trial, and 19,450 breast cancer patients. Additionally, there were no discernible advantages from consuming other non-pro-vitamin A carotenoids [96].

Undoubtedly, one of the most crucial factors in cancer is inflammation. Lycopene is being investigated in several pre-clinical and clinical cancer investigations as one of the most effective anti-inflammatory nutraceuticals. Epidemiological research revealed a negative correlation between the serum level and the development of cancer. Additionally, increasing lycopene intake (from all sources) has been linked to a lower risk of a number of cancers, including breast, lung, prostate, stomach, and ovary [97]. Notably, a recent report has highlighted how the combination of lycopene and enzalutamide, an effective drug for the treatment of castration-resistant prostate cancer (CRPC), can significantly inhibit the proliferation and invasion of CRPC cells in vitro, as well as tumor growth and bone metastasis in vivo. The authors of this research have inferred that the found synergistic anti-tumor effects, probably due to the reduction of AR protein levels through lycopene-mediated inhibition of the AKT/EZH2 pathway, might provide a new approach to improve the efficacy of enzalutamide in CRPC [98].

### 5.4. Skin and Eye Diseases

Because of their UV protection properties, carotenoids are frequently used in cosmeceuticals. Additionally, carotenoids could enhance skin properties [6,99]. A comprehensive evaluation of 11 clinical investigations found that supplementing with 3 to 6 mg per day of astaxanthin for up to 16 weeks enhanced the appearance of the skin (wrinkles) and moisture content [100]. The xanthophylls lutein, zeaxanthin, and meso-zeaxanthin (a metabolic by-product of lutein in the body) accumulate in the human retina fovea and inner plexiform layer as macular melanin, which protects the retinal membrane from the damaging effects of short-wavelength, high-intensity light and enhances visual acuity. Dietary lutein/zeaxanthin has a crucial role to play in lowering the risk of age-related macular degeneration (AMD), according to epidemiological and clinical research [101]. Lycopene can boost skin defense systems by promoting the creation of prostaglandins and phospholipids, which are elements of cell membranes. It inhibits and protects against UVB-induced acute skin damage [102]; photodamage is reduced by suppressing epidermal ornithine decarboxylase and inflammatory reactions, as well as maintaining normal cell proliferation and minimizing DNA damage [103]. According to different research, a rise in plasma isomers of lycopene may alter the stimulation of nuclear hormone receptor signaling pathways and be at least partially accountable for the phenotype of AD [104].

### 5.5. Bone Diseases

A diet high in antioxidants, such as carotenoids, may lessen the age-related decline in physical function and muscle mass. Ingestion of total carotenoids, lutein/zeaxanthin, and lycopene was linked to a greater annual rate change in muscular strength and a quicker walking speed among older individuals (average age of 61 years) [105]. Human osteoblasts and osteoclasts are affected by lycopene in several molecular and cellular ways [100]. It decreased osteoclast differentiation and calcium phosphate resorption but had no effect on cell survival or density [105]. Lycopene also increased osteoblast proliferation (reduced apoptosis) and differentiation [106]. Lycopene supplementation prevented the ovariectomized (OVX)-associated increase in bone turnover by altering serum osteocalcin, cross-linked carboxyterminal telopeptides, bone metabolism, serum N-terminal pro-peptide of type 1 collagen, and urine deoxypyridinoline indicators. Additionally, lycopene treatment decreased osteoclast activity; along with differentiation, the osteoblast, GPx CAT, and SOD activities were also upregulated. In post-menopausal females, daily lycopene ingestion lowers the risk of bone resorption by preventing oxidation [107].

## 6. Security and Toxicology

Regarding therapeutic plants and phytochemicals produced from plants, safety management must be taken seriously. Numerous in vitro and in vivo studies have been carried out on the potential toxicity of lycopene. For instance, it has been demonstrated that the survival of cultured rat cerebellar granule neurons is unaffected by lycopene (up to 10 μM) [108]. Although it appears that carotenoids in some specific conditions at high tissue concentrations may demonstrate a pro-oxidant effect, another study on cultured rat hippocampus neurons showed no discernible harmful effects of lycopene when it was administered to these cells [109]. It is worth emphasizing that various lycopene forms, including those made from synthetic lycopene, tomato extract, and its crystalline extract, are usually regarded as safe when employed in a variety of food items [109]. Lycopene consumption at usual and recommended levels has not been associated with any reported negative effects. It has been determined that daily lycopene consumption is around 123 mg per day [110]. However, there is no perfect amount of lycopene to consume every day; for instance, 6.5 mg per day of lycopene was effective in preventing cancer in males, according to in vivo research [111]. In a different trial, lycopene administration (15 mg per day, for 12 weeks) increased immune function in an elderly population by 28%, increasing natural killer cell activity. Consequently, it appears that for a variety of medical conditions, different lycopene dosages and supplementation periods might be advised. Finally, some epidemiological research implies daily lycopene consumption of 2 mg up to 20 mg daily [112]. Additionally, the mutagenicity of lycopene was evaluated in a series of investigations; the findings suggest that lycopene in its formulated form has no mutagenic effects, however, crystalline lycopene did exhibit some mutagenic activity when it was degraded in the presence of light and air [113].

## 7. Conclusions

An overview of recent literature on carotenoids’ properties, from their structures related to the antioxidant capacity to their potential health benefits associated with the intake by the diet, was presented in this review article. Carotenoids are mostly present in fruits and vegetables, though food of animal origin can be rich in carotenoids, too; alternative sources (e.g., macroalgae and insects) are also becoming increasingly topical.

Apart from the content of these bioactive compounds in food, their bioavailability is very important for beneficial functions in the body, which is strongly influenced by the molecular structure, as well as the interaction with other nutrients (i.e., proteins, lipids, and fibers) and dietary phytochemicals. Prevalently, carotenoid consumption and absorption help to fight oxidative stress and reduce the risk of several chronic diseases, including cardiovascular and neurological disorders, type 2 diabetes, and different types of cancer. This was especially discussed in the case of lycopene. Numerous biological impacts of lycopene have already been demonstrated in studies on animals, cell cultures, and epidemiology. Such results have sparked additional studies into how lycopene and its metabolites contribute to the emergence of chronic disorders.

In conclusion, this review could represent a useful starting point to focus the research on carotenoids as possible ingredients for functional foods and nutraceuticals exploitable in the medicine and cosmetics sectors.

## Figures and Tables

**Figure 1 antioxidants-12-01069-f001:**
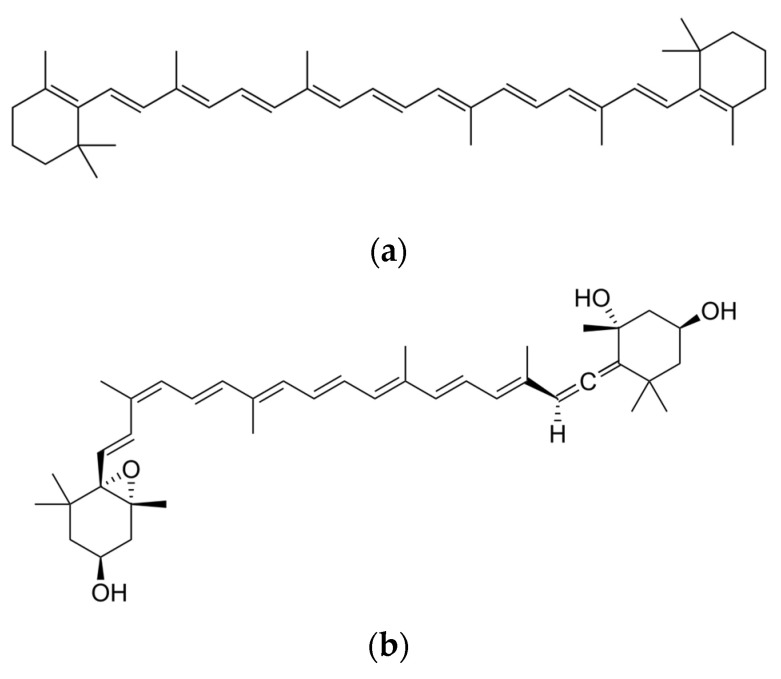
(**a**) Carotene- and (**b**) xanthophyll-like structures.

**Figure 2 antioxidants-12-01069-f002:**
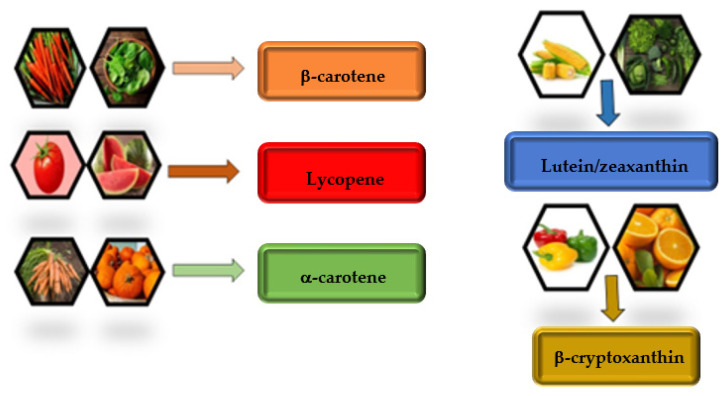
The dietary carotenoids from the major fruits and vegetables.

**Figure 3 antioxidants-12-01069-f003:**
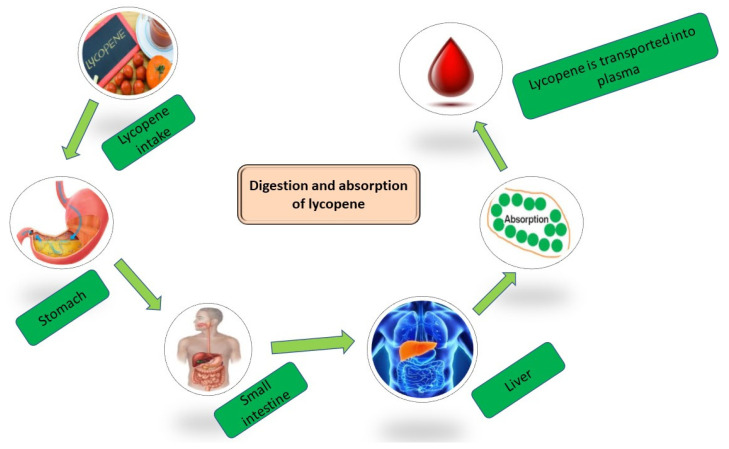
Lycopene absorption from the consumption of food.

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
