# Peer review of "Overview of the Potential Beneficial Effects of Carotenoids on Consumer Health and Well-Being"

_antioxidants, 2023, doi:10.3390/antiox12051069_

Round 1
Reviewer 1 Report
This review provides an overview of research progress on the biochemistry, preventive and therapeutic effects of carotenoids, with particular emphasis on lycopene. As far as the dating of references, it appears to be a review of progress, citing relatively recent ones.
The following is a list of contents that the reviewer would like to see added, and contents that the reviewer believes need to be corrected or confirmed.
1, Bioavailability of lycopene
Lines 329-330, "lycopene is digested and packed into chylomicrons in the small intestine by passive-regulated diffusion."
but please cite references if you have them. Also, there is a report in 2005 (J. Nutr. 135: 2305-2312, 2005.) on the facilitated diffusion of lycopene, but are there any recent findings? If so, please describe them.
2, Bioavailability and functionality of lycopene
Lycopene is known to be cleaved by BCO2, but is there any recent knowledge on the functionality of metabolites such as lycopenal and lycopenoic acid? If so, it should be mentioned.
3,
"5. role of carotenoids in human health and disease: the lycopene case".
From this point onward, the reviewer would like to see a clear distinction made as to whether the functionality of carotenoids (especially lycopene) is based on human studies, animal studies, or cultured cell studies. The reviewer has the impression that many parts of the current version are written with these mixed up. The reviewer thinks most of the reader's interest is probably in the results of human studies.
4, Figure 1 b
Why the structural formula of neoxanthin?
If anything, shouldn't the structural formula of lutein be written?
5, Figure 2
Corn, green vegetables, why only zeaxanthin? What about lutein?
Lutein is more abundant than zeaxanthin, both in food and plasma. At least, both should be included, as in lutein and zeaxanthin.
6, Line 560-562, "For instance, it has been demonstrated that the survival of cultured rat cerebellar granule neurons is unaffected by lycopene (up to 10 M) [102]."
Reference 102 is,
"Carotenoid dietary intakes and plasma concentrations are associated with heel bone ultrasound attenuation and osteoporotic fracture risk in the European Prospective Investigation into Cancer and Nutrition (EPIC)-Norfolk cohort."
but is the No. in the citation correct? It doesn't seem to fit. And what about the other parts? The author should check again.
7. It also says "up to 10 M", but the concentration of lycopen is 10M? Really? 10 M of lycopene is 5370 g/L. Even using THF, the solvent with the highest solubility for carotenoids, the solubility of β-carotene is 10 g/L (JAFC, 40, 431-434, 1992). Concentrations as high as 10 M lycopene seem impossible.
Author Response
Reviewer 1
This review provides an overview of research progress on the biochemistry, preventive and therapeutic effects of carotenoids, with particular emphasis on lycopene. As far as the dating of references, it appears to be a review of progress, citing relatively recent ones. The following is a list of contents that the reviewer would like to see added, and contents that the reviewer believes need to be corrected or confirmed.
Response - We would like to thank the referee for the observation, we have carefully considered your suggestions and edited the manuscript consequently.
Comment - 1. Bioavailability of lycopene: Lines 329-330, "lycopene is digested and packed into chylomicrons in the small intestine by passive-regulated diffusion." but please cite references if you have them. Also, there is a report in 2005 (J. Nutr. 135: 2305-2312, 2005.) on the facilitated diffusion of lycopene, but are there any recent findings? If so, please describe them.
Response - In this regard, the reference related to the sentence "lycopene is digested and packed into chylomicrons in the small intestine by passive-regulated diffusion." is [52]. The bibliographic reference number [52] was previously included.
Following your suggestion, we decided to include the paragraph " It is well known that regular intake of lycopene can prevent chronic diseases such as cardiovascular disease, type 2 diabetes, high blood pressure, kidney disease, bone, skin and eye diseases and cancer. However, thermal processing, light, oxygen, and enzymes in the gastrointestinal tract (GIT) impair the bioaccessibility and bioavailability of ingested lycopene throughout the diet. Several literature studies suggest nanoencapsulation as a potential platform to protect lycopene from external physical agents and enzymatic activity of the human digestive system to lycopene, favoring its bioaccessibility and use as a functional food ingredient for therapeutic treatments” in the revised manuscript to share the findings on facilitated diffusion of lycopene.
The scientific references below have been incorporated:
- Alexandrine During, Harry D. Dawson, Earl H. Har-rison, Carotenoid Transport Is Decreased and Expression of the Lipid Transporters SR-BI, NPC1L1, and ABCA1 Is Downregulated in Caco-2 Cells Treated with Ezetimibe, The Journal of Nutrition, Volume 135, Issue 10, October 2005, Pages 2305–2312, https://doi.org/10.1093/jn/135.10.2305;
- Anwar Ul Alam M, Khatun M, Arif Ul Alam M 2022. Recent Trend of Nanotechnology Applications to Improve Bio-accessibility of Lycopene by Nanocar-rier: A Review. J Food Chem Nanotechnol 8(4): 162-180.
Comment – 2. Bioavailability and functionality of lycopene: Lycopene is known to be cleaved by BCO2, but is there any recent knowledge on the functionality of metabolites such as lycopenal and lycopenoic acid? If so, it should be mentioned.
Response - In full agreement with the Referee the sentences “Moreover, a variety of lycopene isomers, due to physiological cis-trans isomerization occurring in the intestinal epithelium, liver, and stomach, may be found in human plasma. Both trans- and cis-isomers, accounting for up to 50% of the total amount of lycopene, can be cleaved eccentrically by b-carotene oxygenase 2 (BCO2) to give rise lycopenals, lycopenols, and lycopenoic acids, whose beneficial effects were recently highlighted.” were introduced on Lines xxxxx of the revised manuscript.
In addition, references have been added as follow
- Mozos, I.; Stoian, D.; Caraba, A.; Malainer, C.; Horbańczuk, J.; Atanasov, A.G. Lycopene and vascular health. Front. Pharmacol.2018; 9, 521-525;
- Arballo J, Amengual J, Erdman JW Jr. Lycopene: A Critical Review of Digestion, Absorption, Metabolism, and Excretion. Antioxidants (Basel). 2021 Feb 25;10(3):342. doi: 10.3390/antiox10030342.
Comment – 3. "5. role of carotenoids in human health and disease: the lycopene case": From this point onward, the reviewer would like to see a clear distinction made as to whether the functionality of carotenoids (especially lycopene) is based on human studies, animal studies, or cultured cell studies. The reviewer has the impression that many parts of the current version are written with these mixed up. The reviewer thinks most of the reader's interest is probably in the results of human studies.
Response – According to your request, in the revised version, we have grouped together the animal studies as well as cultured cell studies (lines 408-463) and emphasized the clinical studies pertaining to human health (lines 464-670).
Comment - 4, Figure 1 b: Why the structural formula of neoxanthin? If anything, shouldn't the structural formula of lutein be written?
Response – Even though lutein is the most widespread xanthophyll (as reported in the text), we have chosen to depict the structure of neoxanthin simply to display the possible presence of more oxygenated functional groups in the xanthophyll-like structure.
Comment - 5, Figure 2: Corn, green vegetables, why only zeaxanthin? What about lutein? Lutein is more abundant than zeaxanthin, both in food and plasma. At least, both should be included, as in lutein and zeaxanthin.
Response – According to your suggestion, we have edited Figure 2.
Comment - 6, Line 560-562: "For instance, it has been demonstrated that the survival of cultured rat cerebellar granule neurons is unaffected by lycopene (up to 10 M) [102]." Reference 102 is, "Carotenoid dietary intakes and plasma concentrations are associated with heel bone ultrasound attenuation and osteoporotic fracture risk in the European Prospective Investigation into Cancer and Nutrition (EPIC)-Norfolk cohort." but is the No. in the citation correct? It doesn't seem to fit. And what about the other parts? The author should check again.
Response – You are right, we apologize for the mistake. Actually, the correct reference was: “Qu, M.; Nan, X.; Gao, Z.; Guo, B.; Liu, B.; Chen, Z. Protective effects of lycopene against methylmercury-induced neurotoxicity in cultured rat cerebellar granule neurons. Brain Res. 2013, 1540, 92–102”.
Comment - 7. It also says "up to 10 M", but the concentration of lycopen is 10M? Really? 10 M of lycopene is 5370 g/L. Even using THF, the solvent with the highest solubility for carotenoids, the solubility of β-carotene is 10 g/L (JAFC, 40, 431-434, 1992). Concentrations as high as 10 M lycopene seem impossible.
Response – Thank you very much for your careful observation. We again apologize for the typing mistake; it has been corrected in “up to 10 mM” in the revised version (line 685).

Reviewer 2 Report
The argument of the review is quite large and complex. In several cases, the connection between the cited diseases is missed, and this was a good opportunity to avoid a simple report of data already available. Therefore, the main problem is the absence of critical considerations about the reported data, in order to help the reader in understand the quality and utility of the cited publication. Furthermore, some sentences should be clarified.
line 44-45 algae are reported as unicellular organisms and cyanobacteria as multicellular organisms
line 47-48 primary function of carotenoids is the absorption of light, but they are present in high concentration also in no photosynthetic organs and therefore this is not always the primary function, please clarify to avoid confusion in the reader
in several parts of the manuscript, add a comma before as well (lines 55, 180, 348, 438, 475, 479, 503, ...) and before but (line 78)
in line 191 page 4 add fruits to Rosa mosqueta
page 5 line 240 change Bixa Orellana in Bixa orellana
page 8 line 359 change carotenoids. It in carotenoids. It
in Conflict of interest change The author declares in The authors declare
The quality of English could improved in some sentences, since they are too lengthy and in several points the use of the comma could help the reader to understand.
Author Response
Reviewer 2.
The argument of the review is quite large and complex. In several cases, the connection between the cited diseases is missed, and this was a good opportunity to avoid a simple report of data already available. Therefore, the main problem is the absence of critical considerations about the reported data, in order to help the reader in understand the quality and utility of the cited publication. Furthermore, some sentences should be clarified.
We want to thank the reviewer for the suggestion. Consequently, we have tried to improve the quality of our manuscript.
line 44-45 algae are reported as unicellular organisms and cyanobacteria as multicellular organisms.
Because useless, the specifications have been discarded in the revised version.
line 47-48 primary function of carotenoids is the absorption of light, but they are present in high concentration also in no photosynthetic organs and therefore this is not always the primary function, please clarify to avoid confusion in the reader.
We agree with your observation; we meant that the main function of carotenoids is to absorb light in photosynthetic tissues. However, to clarify the concept, we have rephrased it as such: “They protect the photosynthetic apparatus, dissipating excess light energy, and reduce the reactivity of harmful species such as “singlet” oxygen and the excited (i.e., higher energy state) chlorophyll” (lines 47-50, revised version).
In several parts of the manuscript, add a comma before as well (lines 55, 180, 348, 438, 475, 479, 503, ...) and before but (line 78).
Following your suggestion, we have added the commas in the revised version.
In line 191 page 4 add fruits to Rosa mosqueta
Correction done.
page 5 line 240 change Bixa Orellana in Bixa orellana
Correction done.
page 8 line 359 change carotenoids. It in carotenoids. It
Correction done.
in Conflict of interest change The author declares in The authors declare.
Correction done.

Round 2
Reviewer 1 Report
Reviewer 1 has seen the author's response to the comments and the revised manuscript.
The author has made the corrections as noted by Reviewer 1. However, there still appear to be some typos in the manuscript.
1, Fig. 2, "cryptoxanthin" is correct and "criptoxanthin" is wrong.
2, Line 465 "revised" in "been revised as follows." is not clear. Are you confusing it with a sentence in response to a comment?
3, Line 336, reference [56]
Reviewer 1 gave as an example "J. Nutr. 135: 2305-2312, 2005." which the author added as a citation, but this is a 2005 report, so it is a bit old. If possible, reviewer 1 would have preferred to cite a more recent report, but there was no such report? If there were none, then it is fine as it is.
Author Response
In Figure 2 of the original paper the word "criptoxanthin" has been replaced with "cryptoxanthin".
The sentence on Line 465 of the original paper “However, considering the public interest in the preventive and therapeutic action of food constituents on human health and disease, recent evidence of human studies relating to carotenoids effects against chronic inflammation and non-communicable diseases have been revised as follows” have been rewritten as follow in agreement with your suggestion “However, considering the public interest in the preventive and therapeutic action of food constituents on human health and disease, recent evidence of human studies relating to carotenoids effects against chronic inflammation and non-communicable diseases have been revised.”
Regarding the reported observation “Reviewer 1 gave as an example "J. Nutr. 135: 2305-2312, 2005." which the author added as a citation, but this is a 2005 report, so it is a bit old. If possible, reviewer 1 would have preferred to cite a more recent report, but there was no such report? If there were none, then it is fine as it is.” and point 3 “Line 336, reference [56]”, we agree with the Editor that the citation suggested by Referee 1 " J. Nutr. 135: 2305-2312, 2005" is a bit old. In response to Referee 1 the paragraph, " It is well known that regular intake of lycopene can prevent chronic diseases such as cardiovascular disease, type 2 diabetes, high blood pressure, kidney disease, bone, skin and eye diseases and cancer. However, thermal processing, light, oxygen, and enzymes in the gastrointestinal tract (GIT) impair the bioaccessibility and bioavailability of ingested lycopene throughout the diet. Several literature studies suggest nanoencapsulation as a potential platform to protect lycopene from external physical agents and enzymatic activity of the human digestive system to lycopene, favouring its bioaccessibility and use as a functional food ingredient for therapeutic treatments” has already been included in the original manuscript with the correlated bibliographic study by Anwar Ul Alam M et al. published in 2022(Reference [57] in the manuscript).
Although there are additional studies in the literature aimed to identify potentially useful systems to enhance the bioavailability of lycopene, in our opinion, the work of Anwar Ul Alam M et al. (2022) appears more comprehensive and meets the research focus of the current manuscript.

Reviewer 2 Report
The MN was improved as requested. I hope the review will have a significant impact.
Author Response
We thank the reviewer for his/her appreciation.